# The Nero Lucano Pig Breed: Recovery and Variability

**DOI:** 10.3390/ani11051331

**Published:** 2021-05-07

**Authors:** Carmelisa Valluzzi, Andrea Rando, Nicolò P. P. Macciotta, Giustino Gaspa, Paola Di Gregorio

**Affiliations:** 1Scuola di Scienze Agrarie, Forestali, Alimentari ed Ambientali, University of Basilicata, Via dell’Ateneo Lucano 10, 85100 Potenza, Italy; valluzzi.carmelisa@tiscali.it (C.V.); andrea.rando@unibas.it (A.R.); 2Dipartimento di Agraria, Sezione Scienze Zootecniche, University of Sassari, Via De Nicola 9, 07100 Sassari, Italy; macciott@uniss.it; 3Dipartimento di Scienze Agrarie, Forestali e Alimentari, University of Torino, Largo Baccini 2, 10095 Grugliasco (TO), Italy; giustino.gaspa@unito.it

**Keywords:** Nero Lucano pig, Southern Italy, pedigree analysis, inbreeding coefficient, runs of homozygosity (ROH)

## Abstract

**Simple Summary:**

The reduction of biodiversity determines the loss of species and breeds, with the consequent disappearance of production systems, knowledge, cultures and local traditions. The Nero Lucano pig is a native breed of Southern Italy (Basilicata region) recovered, starting from 2001, because of the high quality of its cured meat products. This study gives a picture of the low genetic variability of this breed. Knowledge of individual inbreeding levels allows for planning of interventions to reduce the negative effects of the low effective population size and, then, improve the efficiency of the actual recovery project.

**Abstract:**

The Nero Lucano (NL) pig is a black coat colored breed characterized by a remarkable ability to adapt to the difficult territory and climatic conditions of Basilicata region in Southern Italy. In the second half of the twentieth century, technological innovation, agricultural evolution, new breeding methods and the demand for increasingly lean meat brought the breed almost to extinction. Only in 2001, thanks to local institutions such as: the Basilicata Region, the University of Basilicata, the Regional Breeders Association and the Medio Basento mountain community, the NL pig returned to populate the area with the consequent possibility to appreciate again its specific cured meat products. We analyzed the pedigrees recorded by the breeders and the Illumina Porcine SNP60 BeadChip genotypes in order to obtain the genetic structure of the NL pig. Results evidenced that this population is characterized by long mean generation intervals (up to 3.5 yr), low effective population size (down to 7.2) and high mean inbreeding coefficients (F_MOL_ = 0.53, F_ROH_ = 0.39). This picture highlights the low level of genetic variability and the critical issues to be faced for the complete recovery of this population.

## 1. Introduction

The Nero Lucano (NL) pig is reared in Basilicata and is characterized by a black coat with rough coarse hair-bristles, mean size, long head with straight nose profile, brought-forward ears of medium length, long and thin legs, lean muscles, thick backfat and low number of newborns per delivery.

This breed is well adapted to the mountain habitat and climate conditions of Basilicata. Due to its rusticity, it is reared outdoors where occasional basic shelters can be found. These pigs are able to exploit feed resources available in the environment (such as thistles, carobs, alfalfa, acorns and bulbs) and occasionally receive a feed integration of common grains [1]. Cured products obtained from NL breed reared in these conditions are strongly appreciated by consumers and are sold under the brand “ANTICO SUINO NERO LUCANO”.

In Southern Italy the presence of black pigs can be traced back to 1729 [2]. The pig population consisted of native animals characterized by: black coat, remarkable rusticity and modest growth. They had different denominations in relation to the area they belonged to. The representative morphological types were: the Appulo-Lucano, the Calabro-Lucano, the Cavallino and the Italico [3].

In the second post-war period, technological innovations, evolution of agricultural and breeding methods, demand for lean meat by consumers and increasing production by farmers, have gradually determined the substitution of these native breeds with cosmopolitan ones. In 2001, few subjects (maybe six) showing the typical characteristics of the ancient black pig reared in Basilicata were identified. These subjects, thanks to institutions such as the Basilicata Region, the University of Basilicata, the Regional Breeders Association, the Comunità Montana Medio Basento and a group of breeders, were used to recover the Nero Lucano pig breed.

The aim of this work was to obtain a first picture of the genetic structure of NL pig to be used for the analysis of the evolution of this population. For this purpose, we analyzed the available pedigrees of 226 NL pigs and the results of their genotyping at the 61,565 SNPs of the Illumina Porcine SNP60 BeadChip.

## 2. Materials and Methods

### 2.1. Animals

Animal blood samples were collected for the common veterinary controls of commercial pig farms. On this basis, the animals were managed according to the Directive 98/58/EC, as required by the Directive 2010/63/EU regarding the protection of animals used for scientific purposes. Samples were collected from 226 Nero Lucano pigs (209 sows and 17 boars), born between 2006 and 2014 and belonging to a pedigree of 281 individuals. DNA was isolated by using the NucleoSpin DNA QuickPure kit (Macherey Nagel, Duren, Germany).

### 2.2. Pedigree Analysis

The software ENDOG v.4.8 [4] was used to evaluate pedigree completeness index; number of maximum, complete and equivalent generations; number of ancestors and their contribution to the genetic variability; and inbreeding coefficients (F_PED_). Effective population size (Ne) was estimated via individual increase in inbreeding [5,6]

### 2.3. DNA Analyses

DNA samples were genotyped with the Illumina Porcine SNP60 BeadChip v2. The distribution of SNPs per chromosome was updated according to Illumina PorcineSNP60 v2.0 Manifest File (https://support.illumina.com/downloads/porcinesnp60-v2_product_files.html, accessed on 23 October 2019).

The quality control was accomplished by using PLINK v.1.07 [7] to include samples with a minimum genotyping rate of 95% and SNPs with a minimum 95% call rate. Hardy–Weinberg equilibrium and individual inbreeding coefficients based on molecular information (F_MOL_) were calculated by considering only polymorphic loci (Minimum Allele Frequency, MAF > 0) located on autosomal chromosomes.

The runs of homozygosity (ROH) were obtained by defining a sliding ’window’ of 50 SNPs, a maximum of one heterozygote and one missing call were allowed in the ‘window’, with at least 50 SNPs per ‘window’. Individual inbreeding values based on ROH (F_ROH_) were calculated as F_ROH_ = ΣL_ROH_/L, where ΣL_ROH_ is the total ROH length per individual and L is the autosomal genome length (2265.77 Mb, according to Sscrofa 11.1 chromosome assembly).

Gene location was accomplished by referring to NCBI Sus scrofa Annotation Release 106 (https://www.ncbi.nlm.nih.gov/genome/annotation_euk/Sus_scrofa/106/, accessed on 30 December 2020).

Gene ontology enrichment analysis was performed by using DAVID 6.8 database [8,9] (https://david.ncifcrf.gov/home.jsp, accessed on 30 December 2020).

## 3. Results

### 3.1. Pedigree Analysis

By using the data of the ‘Registro Anagrafico dei Tipi Genetici Autoctoni della Specie Suina’ (Italian Registrar for Autochthonous Swine Breeds) and ENDOG v.4.8 software, it was possible to construct a pedigree of 281 pigs, distributed across 18 farms, characterized by a completeness index decreasing rapidly upstream the grandparents’ generation (Table 1, Appendix A) due to the incomplete registration at the level of third and fourth generations back. In this pedigree, the 281 pigs were traced across three generations, the maximum number of generations traced was five, and the mean equivalent generations value was 1.39.

The average generation interval was 2.88 years, with a maximum for sire–sire interval (3.5 years), and a minimum for dam–dam one (2.32 years) (Table 2).

Analysis of the pedigree showed that the mean inbreeding (F_PED_) and relatedness values were 0.057 and 0.054, respectively. A total of 81 inbred animals, representing 28.82% of the whole pedigree, were characterized by a mean F_PED_ value of 0.197. These individuals were the result of mating between full siblings (13), half siblings (36), and parent–offspring (32). The high amount of parent–offspring mating is likely to be the consequence of both long generation intervals and the free rearing system. Data separated according to the three generations traced are shown in Table 3.

The more than doubled F_PED_ value (0.043→0.109) from generation 1 to generation 2 was determined by a strong increase (0.176→0.628) in the percentage of inbred animals. Such an increase was, however, coupled with a decreased mean F_PED_ value per inbred pig (0.245→0.174). The effective population size (Ne) was very low in both inbred generations (11.5 and 7.2, respectively). In addition, the whole current gene pool was explained by 42 ancestors, with only 8 explaining the 53.27% of the genetic variability (Appendix A).

### 3.2. Microarray Analysis

A total of 226 out of the 281 pigs belonging to the pedigree were genotyped by using the Illumina PorcineSNP60 BeadChip. These individuals were born between November 2006 and January 2014 and according to ENDOG pedigree analysis were distributed in the three generations traced as reported in Table 4.

The distribution of SNPs per chromosome was updated according to the Illumina PorcineSNP60 v2.0 Manifest File by using PLINK (Appendix A). All the animals passed the data quality control (genotyping rate > 95%), and the available 61565 SNPs were reduced to 60600 by the minimum call rate of 95%.

In the analyzed NL pigs, 12.7% of the autosomal SNPs were monomorphic (Minimum Allele Frequency, MAF = 0), whereas 51.38% were characterized by an MAF > 0.05 (Appendix A).

Hardy–Weinberg analysis was accomplished for generations 1 and 2 with the exclusion of generation 0 for which only six individuals were available (Table 4). The percentage of SNPs that were not in the Hardy–Weinberg equilibrium ranged from 5.03% in generation 1 to 6.76% in generation 2. The excess of homozygotes was responsible for the observed Hardy–Weinberg disequilibrium in 57.53% of cases in generation 1 and in 88.57% of cases in generation 2 (Appendix A). These results are in agreement with the threefold increase of inbred animals percentage from generation 1 to generation 2, as evidenced by pedigree analysis (see Table 3).

The inbreeding coefficients based on SNP frequencies (F_MOL_) showed very high average values of 0.53 ± 0.10 and 0.53 ± 0.12 for generations 1 and 2, respectively. Since SNP frequencies estimates on only six samples are insufficiently reliable, F_MOL_ value for generation 0 was not considered. As shown in Figure 1, more than two thirds of pigs are characterized by values higher than 0.50 (about 70% of generation 1 and 73% of generation 2). The distributions of F_MOL_ were very similar in both generations and characterized by similar minimum (0.11 and 0.14) and identical mean (0.53) and maximum values (0.74).

### 3.3. Analysis of Runs of Homozygosity

A search for runs of homozygosity (ROH) in the 226 NL pigs identified a total of 12159 ROH, covering about 38% of the genome (calculated as mean of total ROH length per individual / 2265.77 Mb autosomal genome length). The higher numbers of ROH were observed on SSC1 (1258 for a 29.41% coverage) and SSC14 (1118 for a 41.44% coverage), while the maximum observed coverage (51.20%) was on SSC4. The distribution of ROH was homogeneous among the size classes 2–4 Mb, 4–8 Mb, 8–16 Mb and >16 Mb, while those with a length less than 2 Mb were poorly represented (only 6.74% of total ROH) (Appendix A). In our population, the number of ROH per pig varied from a minimum of 11 to a maximum of 100 with a mean value of 53.8 ± 11.02 (Appendix A), and the total ROH length per animal varied from a minimum of 30.66 Mb to a maximum of 1446.9 Mb with a mean value of 864.67 ± 269.61 Mb.

In order to identify possible NL chromosomal conserved regions, all the ROH were overlapped according to chromosome position defining the maximum shared segment of each ROH. As a result, 1408 consensus match segments shared by a minimum of 30 to a maximum of 214 pigs were obtained. This number was restricted to 171 ROH segments by considering the following thresholds: ROH shared by more than 30% of the analyzed pigs, longer than 500 kb and with at least 20 SNPs. Such segments were distributed on 17 of the 18 autosomes since no ROH satisfying the above- mentioned threshold conditions were observed on SSC3 (Appendix A). The longer common ROH was located on SSC15 (5749.139 kb, shared by 93/226 pigs), whereas the most represented one was located on SSC14, (1856.209 kb, shared by 214/226 pigs).

Gene search in the most represented ROH per each chromosome was accomplished by using Sscrofa11.1 Genome Assembly. The 230 identified genes (Appendix A) were analyzed for gene ontology by using DAVID 6.8 database. According to the results, five genes (CDS1, INPP5J, ITPR3, MTMR3 and PIP4K2A) are involved in the phosphatidylinositol signaling pathway that is engaged in several biological processes such as: membrane trafficking and endosome dynamics, protein trafficking, cell adhesion, polarization and migration [10,11,12]. In addition, six genes (FGF9, KSR1, NF1, PLA2G3, RGL2 and SYNGAP1) are involved in the Ras signaling pathway that is responsible for control of cell proliferation, migration and survival [13]. Furthermore, 11 genes are involved in 10 biological processes (Table 5). Noteworthy, some of these genes are involved in more than one biological process or pathway.

Though methods and parameters for ROH definition differ according to authors, none of the most represented ROH per each chromosome in NL pig overlapped with common ROH identified in other Italian black pig breeds [14,15]. However, when extending the comparison to the most common 171 NL ROH, only four partial ROH overlaps were identified (Table 6). These overlapping regions contain genes whose activities are related to reproduction and production traits such as spermatogenesis (SPATA5) [16], locomotory behavior, meat smell and taste (NOVA1) [14,17], protection from UVB radiation (CALB1) [18], cellular growth and division (DDX10) [19] and lipid composition traits (DECR1) [20].

The inbreeding coefficients based on ROH extension (F_ROH_), obtained for the generations 0, 1 and 2, were characterized by the distributions shown in Figure 2 and by mean values of 0.36 ± 0.12, 0.39 ± 0.11 and 0.38 ± 0.13, respectively.

The correlation coefficients between F_MOL_ and F_ROH_ were very high: 0.97 and 0.98 for generations 1 and 2, respectively. On the contrary, correlations of F_MOL_ and F_ROH_ with F_PED_ were very low varying from a minimum of 0.10 to a maximum of 0.14.

## 4. Discussion

We analyzed the genetic structure of the Nero Lucano (NL) pig, an endangered small population reared in Southern Italy, about ten years after its recovery. At the time of the sample collection the pedigree of this population was characterized by 281 animals distributed across 18 farms. Though results could be affected by the decreasing completeness of the pedigree, this population was characterized by long generation intervals (Table 2), apparently low mean inbreeding value (0.057) and small effective population sizes (Table 3). In particular, the generation intervals in NL pig were, in some cases, more than twice those observed for cosmopolitan breeds and were characterized by higher values for the sire pathways rather than for the dam ones [21,22,23]. Furthermore, the mean F value shows a strong tendency to increase from a generation to another (from 0.043 to 0.109) (Table 3). A marked contribution to the increase of the F value could be due to the high amount of parent–offspring mating, an obvious consequence of both long generation intervals and free rearing system. In addition, the increase of the inbreeding coefficient was, of course, coupled with the reduction of the effective population size (Ne). According to the FAO [24] guidelines for the management of small populations at risk, an acceptable Ne value should be around 50 in order to avoid increase in inbreeding coefficient and for a good population fitness.

Results obtained by the Illumina PorcineSNP60 BeadChip genotyping evidenced that: (i) the percentage of SNPs having an MAF > 0.05 was lower (51.38%) than those observed in three cosmopolitan breeds (Duroc 67.5%, Landrace 80.6% and Yorkshire 80.4%) [25]; (ii) SNPs in Hardy–Weinberg disequilibrium were mainly due to excess of homozygotes; (iii) the inbreeding coefficients based on molecular data (F_MOL_) were very high and comparable with those observed in some small, closed and endangered Spanish (Guadyerbas 0.80 and Torbiscal 0.74) and Chinese (Wuzhishan 0.44) populations [26,27,28]; and (iv) the overlapping of the F_MOL_ curves in the two inbred generations could be the consequence of breeders activities (for example, boar exchanges) to avoid an excessive increase of inbreeding.

The coverage of the NL pig genome with ROH was about 38%, a very high value when compared with an average of 23% reported for cosmopolitan (Duroc, Hampshire, Large White and Landrace) and Asian (Meishan, Jianquhai and Xiang) breeds [29]. Furthermore, short ROH (<2 Mb), determined by recombination events over generations that disrupt long stretches of DNA, were poorly represented (6.74%). These results are in agreement with recent inbreeding events in a small population [30,31]. Of course, it cannot be excluded that the observed low percentage of short ROH was the result of the limited capability of their detection by the used medium-density marker panel [29,32].

A search for genes in the most represented ROH per each chromosome and in the ROH segments overlapping between NL and other Italian black pig breeds allowed the identification of 230 and 13 genes, respectively. Among these, since the NL pigs are characterized by a low number of newborns per delivery (according to our data, mean value of 6.6 ± 1.9) and their cured products are strongly appreciated, two genes, Leukemia Inhibitory Factor (LIF) and 2,4-Dienoyl-CoA Reductase 1 (DECR1), are, according to us, particularly worthy of consideration. In fact, the former shows polymorphisms associated with litter size variation in pig [33,34], and the latter polymorphisms associated with variation in lipid composition traits [20].

As expected, according to the ROH coverage of the genome, the observed high value of F_ROH_ (mean 0.37) is a further indication of the low level of genetic variability of the NL pig population confirming results obtained with both pedigree and molecular data.

The comparison of the inbreeding coefficients, calculated according to pedigree, molecular data and runs of homozygosity, showed the lowest values for F_PED_ and the highest values for F_MOL_. Explanation of the low level of F_PED_ can be found in the low completeness of the pedigree and in the fact that the starting generation is composed of animals with F_PED_ = 0 by default, underestimating the real level of inbreeding. The observed mean F_ROH_ value for generation 0 (0.36) is a clear indication that animals chosen for the recovery of the NL pig were already inbred. Consequently, F_MOL_ and F_ROH_ seem to be more reliable estimates of inbreeding coefficient values. We observed strong correlation coefficients between F_MOL_ and F_ROH_ and low correlation coefficients of F_MOL_ and F_ROH_ with F_PED_ in NL pig. Similar correlation values, based on a very high number of samples belonging to complete pedigrees, were also obtained for Landrace and Large White breeds [35].

All the results presented in this paper highlight the low level of genetic variability of the current NL pig population which, ten years after its recovery, is still at risk. In this contest the possibility of intervention by using individuals already belonging to the NL pig population is confined only to avoid an excessive inbreeding coefficient increase by using, for example, boars showing the lowest F_ROH_ and/or F_MOL_ values and by exchanging them among breeders. In this way, the repeated use of the same boar on strongly genetically related dams of the following generations will be prevented. Furthermore, the use of sires belonging to the “great family of black Southern Italy pig breeds” (for example, Apulo-Calabrese, Nero Siciliano and Sarda), could be considered as a possible extreme approach. In this case, as stated by FAO [24], the aim is to carefully plan the increase of the genetic variability of NL pig preserving the typical characteristics of the breed and its products.

## 5. Conclusions

The results presented in this study highlight the critical issues to be faced for the complete recovery of the Nero Lucano pig breed. Low effective population size, long generation intervals and high inbreeding values depict a population still at risk about ten years after its recovery.

## Figures and Tables

**Figure 1 animals-11-01331-f001:**
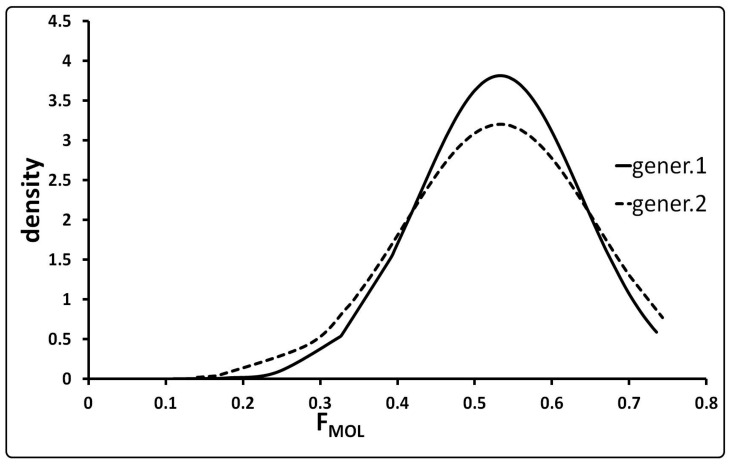
Distribution of the F_MOL_ (inbreeding coefficient using molecular polymorphisms) values in generations 1 and 2 of Nero Lucano pig.

**Figure 2 animals-11-01331-f002:**
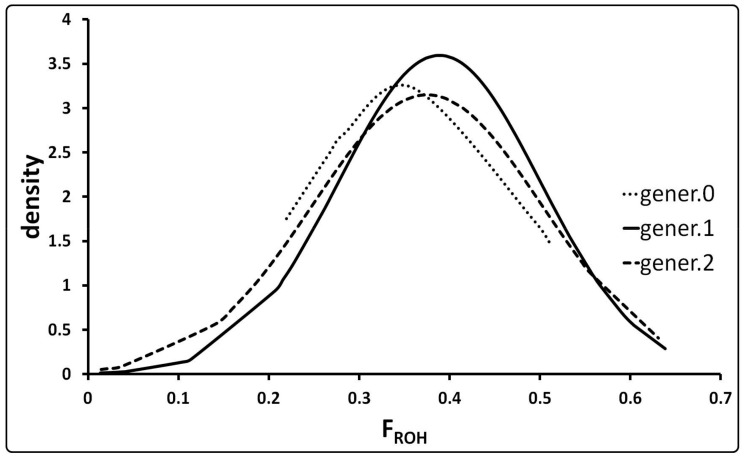
Distribution of the F_ROH_ (inbreeding coefficient using Runs of Homozygosity) values in the three generations of Nero Lucano pig.

**Table 1 animals-11-01331-t001:** Pedigree completeness index for the known generations in Nero Lucano pigs.

Generation	Completeness Index
Parents	0.851
Grandparents	0.461
Great-grandparents	0.072
gg-grandparents	0.003

**Table 2 animals-11-01331-t002:** Average generation intervals in Nero Lucano pigs.

Interval	*N*	Years ± SD
sire–sire	13	3.503 ± 1.469
sire–dam	62	3.250 ± 1.320
dam–sire	13	3.187 ± 0.836
dam–dam	62	2.328 ± 0.778
Total	150	2.886 ± 1.190

**Table 3 animals-11-01331-t003:** Inbreeding coefficient, relatedness and effective population size of the three generations traced in Nero Lucano pigs.

Generation	*N* Pigs	Mean F	% Inbred	Mean Ffor Inbred	Mean Relat.	Eff.Pop. Size
0	42	0	──	──	0.0238	──
1	153	0.043	17.6	0.245	0.0565	11.5
2	86	0.109	62.8	0.174	0.0643	7.2

**Table 4 animals-11-01331-t004:** Number of Nero Lucano pig DNA samples analyzed with Illumina PorcineSNP60 BeadChip within each generation.

Generation	*N* Pigs	
	Pedigree	DNA Samples
0	42	6
1	153	132
2	86	85

**Table 5 animals-11-01331-t005:** Genes associated in biological processes by DAVID software.

GO Term	Biological Process	Genes
GO:0042992	negative regulation of transcription factor import into nucleus	RAB23, NF1
GO:0046888	negative regulation of hormone secretion	LIF, OSM
GO:0007265	Ras protein signal transduction	KSR1, NF1, SYNGAP1
GO:0007260	tyrosine phosphorylation of Stat3 protein	LIF, OSM
GO:0043410	positive regulation of MAPK cascade	KSR1, LIF, OSM
GO:2000786	positive regulation of autophagosome assembly	KIAA1324, PIP4K2A
GO:0045835	negative regulation of meiotic nuclear division	LIF, OSM
GO:0001675	acrosome assembly	TMF1, PLA2G3
GO:0048711	positive regulation of astrocyte differentiation	BIN1, LIF
GO:0048169	regulation of long-term neuronal synaptic plasticity	NF1, SYNGAP1

**Table 6 animals-11-01331-t006:** Genes located in common ROH shared by Nero Lucano and Italian black pigs.

SSC	*N* ^1^	Shared ROH Region	Breed ^2^	Reference	NCBI Genes in Shared Region
4	77	45.15–46.82	AC	[15]	CALB1, DECR1, TMEM64, NBN, NECAB1, C4H8orf88, OTUD6B, SLC26A7, TMEM55A
7	141	72.73–73.13	CA	[14]	NOVA1
8	136	100.93–101.22	AC, CS	[14]	SPATA5
9	165	37.15–37.81	AC	[15]	DDX10, C9H11orf87

^1^ number of NL pigs that show the ROH. ^2^ AC = Appulo Calabrese, CA = Casertana, CS = Cinta Senese.

## Data Availability

The data analyzed during the current study are available from the corresponding author on reasonable request.

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
