# Peer review of "The Nero Lucano Pig Breed: Recovery and Variability"

_animals, 2021, doi:10.3390/ani11051331_

Round 1

Reviewer 1 Report

The revised version looks good now.

Author Response

Thank you for reviewing the manuscript.

Reviewer 2 Report

It is true that the pedigree is incomplete but not shallow. These are all the available data from the Registrar for Autochthonous Swine Breeds and it is stated in the first sentence of  RESULTS.  However Endog software gives the results shown in tables and text and these results, together with molecular data, contribute to give the most complete picture of the genetic variability of the Nero Lucano pig. Just to remind that inbreeding can be detected only if both maternal and paternal ancestries are known. Thank you for the reminder, in fact in table 3 only 17.6% of generation 1 and 62.8% of generation 2 were identified by ENDOG as inbreed individuals. Please, check paragraph 2.4 of ENDOG v4.8 manual. Individual inbreeding coefficients from a given number of founders or ancestors following Lacy et al. (1996) can be also calculated. REPLY: I understand that you use all the pedigree data available. However, because the particularly low pedigree completeness discussion / interpretation should be addressed mainly to molecular inbreeding.  We know that “Individual inbreeding coefficients from a given number of founders or ancestors following Lacy et al. (1996) can be also calculated”, but from what is written in your manuscript you have not used this method.

Table 2: non necessary. Rejected, in general, all papers where generation intervals are considered show a table with the four generation ways. REPLY: It was a suggestion; the information on all pathways is, at my advice, of little interest.

I do not understand the term of “complete generations” and the” three complete generations (LL 102 and 104 and 115 and 127)”. No one generation is complete in the breed pedigree: parents (85%), grandparents (46%), great-grand parents (7%), etc. Rejected: you cannot determine if there are complete generation in this way for species in which generation are not separated. REPLY: You cannot REJECT unless you provide scientific literature that what you say on overlapping generations is true. At my knowledge, completeness is computed following “MacCluer J, Boyce B, Dyke L, Weitzkamp D, Pfenning A, Parsons C, 1983. Inbreeding and pedigree structure in Standardbred horses. J. Hered. 74: 394-399.” See page 395. As you show in your table and in the Figure no one generation is complete. Text should be changed accordingly: OK, you decide to use three generations although the completeness of the third is minimal (7%).

ENDOG computes the “equivalent complete generations” that differs from “complete generations”. Text should be modified accordingly. Rejected: Endog computes “J_GenMax (which is the maximum number of generations traced), J_GenCom (which is the number of full generations traced) and J_GenEqu (which is the equivalent complete generations)”. Lane 103-104. We chose to analyze population according to full (called also complete, see fig.7 of ENDOG manual) generations traced in order to have classes with an acceptable number of individuals. REPLY: Please read the manual “3.2.8. The PorG and PorC Tables These tables are produced by using the Inbreeding per Generations submenu and are basically the same but showing the information by number of full generations traced (PorC) or by maximum number of generations traced (PorG). Names of fields are: J_GenMax (or J_GenCom) which is the number of generations traced (not full generations, at my knowledge), N (individuals per generation), F (average inbreeding), POR (percent inbred individuals), FP (average inbreeding for inbred individuals, AR (mean average relatedness) and, finally, Ne (effective size) if Ft>Ft-1. If different, please proved scientific literature supporting it.

LL 265-266: please consider that, by reducing generation interval, rate of inbreeding per year will increase! True, however “a repeated use of the same boar on dams of the following generations”, that is parent-offspring matings, “obvious consequence of both long generation intervals and free rearing system”, is responsible for a huge increase of inbreeding coefficient. REPLY: Preserving genetic variation in a small population is generally addressed to average coancestry (or average inbreeding) which is function of effective population size and generation interval (inversely) under random mating. Inbreeding coefficients due to some matings between close relatives do not have much meaning.

LL271-275: preferably, to be moved to Conclusions. Rejected: this is not a conclusion, but a further possibility of intervention to be used together with the previously mentioned one. REPLY: It was only a suggestion, does not need rejection.

Author Response

It is true that the pedigree is incomplete but not shallow. These are all the available data from the Registrar for Autochthonous Swine Breeds and it is stated in the first sentence of  RESULTS.  However Endog software gives the results shown in tables and text and these results, together with molecular data, contribute to give the most complete picture of the genetic variability of the Nero Lucano pig. Just to remind that inbreeding can be detected only if both maternal and paternal ancestries are known. Thank you for the reminder, in fact in table 3 only 17.6% of generation 1 and 62.8% of generation 2 were identified by ENDOG as inbreed individuals. Please, check paragraph 2.4 of ENDOG v4.8 manual. Individual inbreeding coefficients from a given number of founders or ancestors following Lacy et al. (1996) can be also calculated. REPLY: I understand that you use all the pedigree data available. However, because the particularly low pedigree completeness discussion / interpretation should be addressed mainly to molecular inbreeding.

We analyzed the genetic structure of the Nero Lucano pig by using pedigree and molecular analyses based on SNPs and ROH and more than 60% of discussion/interpretation is addressed to molecular results. Other authors using less complete pedigrees in cattle published their results on pedigree analyses (Gutierrez et al., 2003. Genet.Sel.Evol 35:46-63). This reference is also indicated in Endog v.4.8 manual.

 We know that “Individual inbreeding coefficients from a given number of founders or ancestors following Lacy et al. (1996) can be also calculated”, but from what is written in your manuscript you have not used this method.

In fact we did not use a method, we used ENDOG software which uses different methods, formulas and references necessary for the analysis of the pedigree. We correctly cited authors of ENDOG software in order to avoid to cite dozens of references.

Table 2: non necessary. Rejected, in general, all papers where generation intervals are considered show a table with the four generation ways. REPLY: It was a suggestion; the information on all pathways is, at my advice, of little interest.

If it's a suggestion, we prefer to keep the table

I do not understand the term of “complete generations” and the” three complete generations (LL 102 and 104 and 115 and 127)”. No one generation is complete in the breed pedigree: parents (85%), grandparents (46%), great-grand parents (7%), etc. Rejected: you cannot determine if there are complete generation in this way for species in which generation are not separatedREPLY: You cannot REJECT unless you provide scientific literature that what you say on overlapping generations is true. At my knowledge, completeness is computed following “MacCluer J, Boyce B, Dyke L, Weitzkamp D, Pfenning A, Parsons C, 1983. Inbreeding and pedigree structure in Standardbred horses. J. Hered. 74: 394-399.” See page 395. As you show in your table and in the Figure no one generation is complete. Text should be changed accordingly: OK, you decide to use three generations although the completeness of the third is minimal (7%).

Changed accordingly. MacCluer et al. 1983 is the reference given by Endog v.4.8 manual for the estimate of pedigree completeness

ENDOG computes the “equivalent complete generations” that differs from “complete generations”. Text should be modified accordingly. Rejected: Endog computes “J_GenMax (which is the maximum number of generations traced), J_GenCom (which is the number of full generations traced) and J_GenEqu (which is the equivalent complete generations)”. Lane 103-104. We chose to analyze population according to full (called also complete, see fig.7 of ENDOG manual) generations traced in order to have classes with an acceptable number of individuals. REPLY: Please read the manual “3.2.8. The PorG and PorC Tables These tables are produced by using the Inbreeding per Generations submenu and are basically the same but showing the information by number of full generations traced (PorC) or by maximum number of generations traced (PorG). Names of fields are: J_GenMax (or J_GenCom) which is the number of generations traced (not full generations, at my knowledge), N (individuals per generation), F (average inbreeding), POR (percent inbred individuals), FP (average inbreeding for inbred individuals, AR (mean average relatedness) and, finally, Ne (effective size) if Ft>Ft-1. If different, please proved scientific literature supporting it.

Changed accordingly

LL 265-266: please consider that, by reducing generation interval, rate of inbreeding per year will increase! True, however “a repeated use of the same boar on dams of the following generations”, that is parent-offspring matings, “obvious consequence of both long generation intervals and free rearing system”, is responsible for a huge increase of inbreeding coefficientREPLY: Preserving genetic variation in a small population is generally addressed to average coancestry (or average inbreeding) which is function of effective population size and generation interval (inversely) under random mating. Inbreeding coefficients due to some matings between close relatives do not have much meaning.

The wrong period was changed. However, in the analyzed population, matings between parent-offspring (32) and full sibs (13) (see text) represent the 55% of matings between relatives.

LL271-275: preferably, to be moved to Conclusions. Rejected: this is not a conclusion, but a further possibility of intervention to be used together with the previously mentioned one. REPLY: It was only a suggestion, does not need rejection.

Round 2

Reviewer 2 Report

Dear Authors, you have taken into account most suggestions. However, on the problem of pedigree completeness I do not understand why you wrote "changed accordingly" but you just changed the term "complete" with the term "full": these terms are synonymous!  (https://www.collinsdictionary.com/dictionary/english-thesaurus/full)

Please, I strongly advise you to remove the term full. None of the generations is full/complete, according to MacCluer et al.

Author Response

The term “full” was removed from the text according to your advice

This manuscript is a resubmission of an earlier submission. The following is a list of the peer review reports and author responses from that submission.

Round 1

Reviewer 1 Report

Manuscript: The Nero Lucano pig breed: recovery and variability

In the current manuscript, the authors tried to improve the inbreeding plans for the low genetic viability of Nero Lucano pig and to reduce the negative effects of the low effective population size. They found that the population of Nero Lucano pig displays long mean generation intervals, low effective population size, and high mean inbreeding coefficients. This work provided basic instruction for the well-done recovery of this population. Some questions should be addressed before publication.

Major questions:

  1. In Table 2, What’s the major difference between Sire and Dam in genetic? Why do two populations show different interval years?
  2. In Table 3, Why does generation 2 present low numbers?

Author Response

  1. In Table 2, What’s the major difference between Sire and Dam in genetic? Why do two populations show different interval years? In general, for animal species reared for economic reasons, male reproducer is called sire and female reproducer is called dam. Contrary to what observed in cosmopolitan breeds, Nero Lucano boars, compared with sows, are characterized by a longer reproductive life
  2. In Table 3, Why does generation 2 present low numbers? Number of pigs: these were the data available from the breeder association and are probably determined by the fact that breeders sold part of piglets without registration, The low Mean F for inbreed individuals is due to a different distribution of type of matings between relatives in the two generations

Reviewer 2 Report

line73-76: Please provide more details on how these parameters were estimated such as pedigree completeness and equivalent generations, there should be formulas for these parameter estimations.

line151: similar minimum, mean, and maximum values are not the reason for the similar distribution of F in two generations but a description.

Figure 1 and 2: give a scale for the x-axis

line231: are they comparable? What is the population size of the Spanish and Chinese populations? The inbreeding level can be affected largely by the size and breeding strategy, please provide more details for the two populations. 

line236: again, provide more details about the referred European and Asian breeds such as the population size.

line245: how low is the number of newborns per littler for the NL breed

line252: F_ped level was much lower than F_mol and is not consistent with the F_roh.

line255: note that F_ped is based on different assumptions, which assume a neutral locus that is not linked to any QTL, ignore the historic inbreeding, and depend on the accuracy of pedigree. 

line 258: remove "cut"

Author Response

line73-76: Please provide more details on how these parameters were estimated such as pedigree completeness and equivalent generations, there should be formulas for these parameter estimations. All the parameters were estimated according to the formulas indicated in the ENDOGv.4.8 MANUAL and in the accompanying references reported in line 73 and 77. Consequently, the insertion of the formulas appears strongly redundant.

line151: similar minimum, mean, and maximum values are not the reason for the similar distribution of F in two generations but a description. Changed accordingly.

Figure 1 and 2: give a scale for the x-axis we are sorry but the figures in the original manuscript contained both axes.

line231: are they comparable? What is the population size of the Spanish and Chinese populations? The inbreeding level can be affected largely by the size and breeding strategy, please provide more details for the two populations. Changed accordingly.

line236: again, provide more details about the referred European and Asian breeds such as the population size. Changed accordingly.

line245: how low is the number of newborns per littler for the NL breed Results given.

line252: F_ped level was much lower than F_mol and is not consistent with the F_roh. Rejected: there is no reference to FPED in this line. “Pedigree and molecular data” are referred to the different results obtained from pedigree and molecular analyses

line255: note that F_ped is based on different assumptions, which assume a neutral locus that is not linked to any QTL, ignore the historic inbreeding, and depend on the accuracy of pedigree. Rejected: in this line we give only a possible explanation of why FPED is low based on the data we collected, that is the pedigree is incomplete and no one knows the historic inbreeding of the founders (FPED = 0 by default).

line 258: remove "cut" Changed accordingly.

Reviewer 3 Report

Dear Authors,

the breed pedigree is particularly shallow and incomplete, it cannot provide much information. Just to remind that inbreeding can be detected only if both maternal and paternal ancestries are known. L217-218: due to poor pedigrees, the sentence has no much meaning.

  1. 69: why partial?

Table 2: non necessary.

LL : 73-74 “completeness index” reported in Methods but not used in Results.

How effective size has been computed? (ENDOC provides several different computations). Moreover, please use only the references utilised (L.73)

I do not understand the term of “complete generations” and the” three complete generations (LL 102 and 104 and 115 and 127)”. No one generation is complete in the breed pedigree: parents (85%), grandparents (46%), great-grand parents (7%), etc. ENDOG computes the “equivalent complete generations” that differs from “complete generations”. Text should be modified accordingly.

LL 226-227: Comparison in MAF with other local breeds (Italian or not) is possible?

LL 265-266: please consider that, by reducing generation interval, rate of inbreeding per year will increase!

LL271-275: preferably, to be moved to Conclusions.

Author Response

the breed pedigree is particularly shallow and incomplete, it cannot provide much information. It is true that the pedigree is incomplete but not shallow. These are all the available data from the Registrar for Autochthonous Swine Breeds and it is stated in the first sentence of  RESULTS.  However Endog software gives the results shown in tables and text and these results, together with molecular data, contribute to give the most complete picture of the genetic variability of the Nero Lucano pig.

Just to remind that inbreeding can be detected only if both maternal and paternal ancestries are known. Thank you for the reminder, in fact in table 3 only 17.6% of generation 1 and 62.8% of generation 2 were identified by ENDOG as inbreed individuals. Please, check paragraph 2.4 of ENDOG v4.8 manual. Individual inbreeding coefficients from a given number of founders or ancestors following Lacy et al. (1996) can be also calculated.

 L217-218: due to poor pedigrees, the sentence has no much meaning. Changed.

  1. 69: why partial? Changed accordingly.

Table 2: non necessary. Rejected, in general, all papers where generation intervals are considered show a table with the four generation ways.

LL : 73-74 “completeness index” reported in Methods but not used in Results. We omitted “index” in line 101 and table 1. Changed accordingly.

How effective size has been computed? (ENDOC provides several different computations). Moreover, please use only the references utilised (L.73) Changed accordingly.

I do not understand the term of “complete generations” and the” three complete generations (LL 102 and 104 and 115 and 127)”. No one generation is complete in the breed pedigree: parents (85%), grandparents (46%), great-grand parents (7%), etc. Rejected: you cannot determine if there are complete generation in this way for species in which generation are not separated.

ENDOG computes the “equivalent complete generations” that differs from “complete generations”. Text should be modified accordingly. Rejected: Endog computes “J_GenMax (which is the maximum number of generations traced), J_GenCom (which is the number of full generations traced) and J_GenEqu (which is the equivalent complete generations)”. Lane 103-104. We chose to analyze population according to full (called also complete, see fig.7 of ENDOG manual) generations traced in order to have classes with an acceptable number of individuals.

LL 226-227: Comparison in MAF with other local breeds (Italian or not) is possible? Reference n.14 deals with local breeds, however the comparison is impossible for the different pruning approach used.

LL 265-266: please consider that, by reducing generation interval, rate of inbreeding per year will increase! True, however “a repeated use of the same boar on dams of the following generations”, that is parent-offspring matings, “obvious consequence of both long generation intervals and free rearing system”, is responsible for a huge increase of inbreeding coefficient.

LL271-275: preferably, to be moved to Conclusions. Rejected: this is not a conclusion, but a further possibility of intervention to be used together with the previously mentioned one.